# Russian Registry of Idiopathic Pulmonary Fibrosis: Clinical Features, Treatment Management, and Outcomes

**DOI:** 10.3390/life13020435

**Published:** 2023-02-03

**Authors:** Svetlana Chikina, Alexander Cherniak, Zamira Merzhoeva, Igor Tyurin, Natalia Trushenko, Anna Proshkina, Kirill Ataman, Sergey Avdeev

**Affiliations:** 1Department of Pulmonology, Sechenov First Moscow State Medical University (Sechenov University), 8, Build.2, Trubetskaya Str., Moscow 119991, Russia; 2Federal Pulmonology Research Institute, Federal Medical and Biological Agency of Russia, 28, Orehovyi Bul., Moscow 115682, Russia; 3Russian Medical Academy for Postgraduate Education, 2/1, Build.1, Barrikadnaya Str., Moscow 125993, Russia

**Keywords:** idiopathic pulmonary fibrosis, registry, antifibrotic agents, multidisciplinary discussion

## Abstract

**Simple Summary:**

Idiopathic pulmonary fibrosis (IPF) is a chronic progressive lung disease of unknown etiology characterized by the rapid development of respiratory failure and a poor outcome. No effective treatment is available worldwide to stop the progression of this disease and to recover the fibrotic lesions in the lungs. Registries of such patients help to better understand this disease, to make a diagnosis as early as possible, and to find the optimal treatment. We present the results of the first experience in our country of integrating the findings of patients with IPF in a registry. This experience highlighted the difficulties in making the diagnosis of IPF and choosing treatment options. We also compared the Russian population of patients with IPF with registries of similar patients in other countries. This allows further perspectives in management of this disease to be defined.

**Abstract:**

A registry of patients with idiopathic pulmonary fibrosis (IPF) was founded in Russia in 2016. The aim of this study was to analyze the demographic, clinical, functional, radiological, and morphological data of the patients included in this registry. Methods. This was a prospective multicenter, observational, non-interventional study. Patients’ risk factors, demographics, clinical data, results of high-resolution computed tomography (HRCT) of the chest and pulmonary function testing, and lung tissue biopsy findings were analyzed. We also analyzed the exercise tolerance (6-min walking test) of patients, serological markers of systemic connective tissue diseases, treatment, clinical course, and outcomes of the disease. Multidisciplinary discussion (MDD) was used as needed. Results. One thousand three hundred and fifty-three patients were included in the registry from 2016 to 2020. The mean age was 64.4 ± 10.7 years, most patients were active smokers or ex-smokers. Antifibrotic therapy was administered to 90 of 948 patients (9.5%). Since starting the registry in 2016, the incidences of IPF have increased and the time period from manifestation of the disease to making the diagnosis has shortened, the number of patients on antifibrotic therapy has increased and the number of patients taking systemic steroids decreased. Conclusion. The registry of patients with IPF was helpful to improve IPF diagnosis and to implement antifibrotic agents in clinical practice. Further analysis of the clinical course and prognostic markers of IPF in the Russian population is needed. An analysis of the long-term efficacy of antifibrotic therapy in this population is also important.

## 1. Introduction

Idiopathic pulmonary fibrosis (IPF) is a specific variant of chronic progressive interstitial pneumonia of unknown etiology. IPF occurs prevalently in elderly patients and is characterized by the rapid development of progressive respiratory failure and by a poor outcome [1,2,3]. IPF is of great importance due to its irreversible progressive course resulting in patients’ disability and death.

Recently, two antifibrotic drugs, pirfenidone and nintedanib, have been approved for the treatment of IPF based on large randomized clinical trials [4,5,6,7]. In addition to conducting clinical trials to evaluate the efficacy and safety of new treatments, observational studies from real clinical practice are also needed to understand the natural course of the disease and patients’ responses to therapy [8]. Data from real patient registries may be more representative of clinical practice than results from clinical trials. The establishment of patient registries has enabled the collection of data from cohorts of patients with IPF with a broader spectrum of disease severity and comorbidities managed in clinical practice, thereby providing a better understanding of the real world behavior and impact of IPF [9]. In recent years, a number of IPF registries have been established to describe the epidemiology, the natural course, and the clinical management of real patient populations with IPF [10,11,12,13,14,15,16,17,18,19,20,21,22,23]. Data from patient registries have already improved knowledge of the clinical course and impact of IPF, and of diagnostic and treatment practices. The national registries have the potential to optimize patient management and improve the diagnosis of IPF, including earlier diagnosis. An early diagnosis, and the initiation of appropriate evidence-based treatment, could improve the poor prognosis of IPF [24].

In 2016, the Russian Respiratory Society published national clinical guidelines on the diagnosis and treatment of IPF [25]. In the same year, a Russian registry of patients with IPF was started. The aim of this study was to analyze the epidemiology, risk factors, clinical course, and therapy of IPF in the Russian population in a real clinical practice.

## 2. Materials and Methods

This was a nationwide prospective multicenter, observational, non-interventional study. Patients who entered the registry before the 4 April 2020 were included in the analysis. To this date, 291 physicians from 140 medical facilities of the Russian Federation had participated in the registry. Physicians managing IPF patients were eligible for participation in the study, considering their qualifications and their past participation and experience in similar studies. After receiving a response about readiness to participate in the project, an initial general meeting was held to discuss the study protocol. Also, during the first two years of the project, regular discussions and reviews of selected clinical cases were held.

Patients. Physicians submitted to the registry the following information for each patient: demographic data (age, gender, height, weight), the date of manifestation of the disease and the date of its diagnosis; the risk factors for IPF (smoking, environmental exposures, genetic factors, etc.); comorbidities; the patient’s symptoms and physical examination findings at the moment of inclusion to the registry (respiratory rate (RR), heart beat rate (HBR), blood pressure (BP), oxygen saturation (SpO2), lung auscultation findings, peripheral oedema, cyanosis, clubbed fingers); pulmonary function parameters; HRCT findings; laboratory findings, including blood markers of connective tissue diseases (CTD); 6-min walk test (6-MWT) results and histopathological findings. Dyspnea on daily life activities was evaluated using the Medical Research Council (MRC) scale [26]. Dyspnea before and after the 6-MWT was measured using the Borg’s scale [27]. Physicians also reported the patient’s treatment taken before inclusion in the registry.

In subsequent years, physicians added to the registry follow-up information including, change in clinical status of the patient, serial measurement of pulmonary function and serial HRCT, and change in treatment, including lung transplantation. For patients who had died, the date and cause of death were reported.

HRCT was performed in accordance with standard requirements for the diagnosis of IPF [23]. HRCT scans were referred to a radiologist with experience of diagnosis of interstitial lung diseases (ILDs). Lung tissue biopsy specimens were also referred to a pathologist with experience of the diagnosis of ILDs. The total information on each patient was subsequently analyzed by a clinician with experience of the diagnosis and differential diagnosis of ILDs. In the case of diagnostic uncertainty, i.e., inconsistency or a lack of clinical, radiological, and histopathological findings, a central multidisciplinary discussion (MDD) review was used.

Bearing in mind that the first patients were included in the registry in 2016, there were some differences in the terminology in our registry. According to the ATS/ERS/JRS/ALAT Evidence-based Guidelines for Diagnosis and Management of IPF, 2011, HRCT patterns were classified as usual interstitial pneumonia (UIP), possible UIP, and inconsistent with UIP pattern; and histopathological features were defined as UIP pattern, probable UIP, possible UIP, and inconsistent with UIP pattern [2,3]. This classification was changed in 2018; according to the updated Official ATS/ERS/JRS/ALAT Clinical Practice Guideline for diagnosis of IPF, both HRCT and histopathological patterns were revised and defined as UIP, probable UIP, undetermined for UIP, and alternative diagnosis [2].

All patients were stratified to subgroups according to the length of the disease > 6 or ≤ 6 months. The length of the disease was estimated from the date of the first manifestation of symptoms to the date of entering the registry.

Diagnosis of IPF. The diagnoses of IPF were confirmed by pulmonologists and/or multidisciplinary discussion (MDD) conferences with ILD clinicians, radiologists, and lung pathologists to establish diagnoses using a standardized approach based on approved guidelines. IPF was diagnosed in accordance with international and national guidelines [2,3,25,28]. In patients with a high clinical probability of IPF, and HRCT pattern not consistent with UIP, in whom a biopsy was not available, a working diagnosis of IPF was made [8]. Unclassifiable idiopathic interstitial pneumonia (IIP) was defined as the absence of a confident diagnosis, using an MDD as the standard for ILDs diagnosis [29].

*Statistical analysis*. The analysis included data available to the 4 April 2020. Data were analyzed using methods of descriptive statistics. For discrete variables, the mean and standard deviation (M ± SD) were calculated. For continuous variables, the median (Me) and interquartile range (IQR) were calculated. Subgroups of patients were compared using Student’s test (for normally distributed variables) or Mann–Whitney test (for abnormally distributed variables).

## 3. Results

### 3.1. Patients’ Characteristics

The first patient entered the registry on the 19 December 2016. To the 4 April 2020, the registry included 1353 patients (42.5% males); the mean age was 64.4 ± 10.7 years; the mean body mass index (BMI) was 27.6 ± 4.7 kg/m^2^. The median time from the first symptom’s manifestation to the time of diagnosis was 10.5 (3.0–27.9) months (Figure 1).

Of 1007 patients with information on risk factors available, 450 patients were ex-smokers (44.7%), 127 patients were active smokers at the time of inclusion (12.6%), and 430 patients had never smoked (42.7%). The mean smoking history was 29.3 ± 17.7 pack-years. Hazardous environmental exposures (inorganic dust, livestock, birds, mold, etc.) were reported for 436 patients (43.3%); 15 patients had a family history of IPF (1.5%).

Information on comorbidity was available for 1344 patients. The most frequent comorbidities were arterial hypertension (56.5%), ischemic heart disease (34.0%), and gastroesophageal reflux disease (15.8%) (Figure 2).

#### 3.1.1. Clinical Data

The most frequently reported symptoms and signs at baseline are shown in Figure 2, and included dyspnea (96.4%), crackles (93.8%), a dry cough (79.1%), weakness (57.9%), and fatigue (44.9%) (Figure 3). The MRC dyspnea was graded as 2 or 3 by most patients (*n* = 448 (34.4%) and *n* = 406 (31.1%), respectively); 69 patients (5.3%) experienced dyspnea on vigorous exertion (MRC grade 0), and 179 patients (13.7%) reported MRC dyspnea grade 4.

Pulse oximetry measurements were available for 1284 patients; the median SpO2 at rest while breathing by room air was 94% (91–96%), the minimum SpO2 at rest was 60%.

Blood eosinophils were > 300 cells/µL in 71 of 602 patients (11.8%) who had blood cell count data. Serological biomarkers of CTD were measured in 510 patients; at least one biomarker was positive in 9.2% of the patients.

#### 3.1.2. Functional Measurements

Pulmonary function tests were performed in 921 of 1353 patients. Single-breath diffusion capacity of the lungs for carbon monoxide was measured in 386 of 1353 patients. The mean FVC at baseline was 69.7 ± 22.3% pred. The mean TLC at baseline was 71.5 ± 18.8% pred. The mean DLCO at baseline was 46.2 ± 17.1% pred. The distribution of FVC and DLCO data at baseline are shown in Figure 4 and Figure 5.

A 6-MWT was performed by 526 patients at the point of inclusion to the registry. The median distance walked was 300 (194–405) m; 46.8% of the patients walked < 300 m in the 6 min. The median Borg’s dyspnea score was 2 (2–3) at baseline, and 5 (3–6) at the end of 6-MWT.

#### 3.1.3. HRCT Patterns

HRCT was performed on 1319 of 1353 patients. To the index data, 501 HRCT scans were referred to an experienced radiologist. A UIP pattern was diagnosed in 84 of 501 patients (17.5%); a probable/possible UIP pattern was diagnosed in 171 of 501 patients (34.2%), and an indeterminate HRCT pattern was diagnosed in 22 patients (4.5%). Experienced radiologists detected HRCT signs of alternative diseases (cysts, nodules, prevalent ground-glass opacification, lung tissue consolidation, and peribronchial/perilymphatic or upper-mid lung predominant distribution) in 173 of 501 patients (34.5%). An HRCT pattern suggestive of chronic hypersensitivity pneumonitis (HP) was found in 63 patients, and an HRCT pattern of non-specific interstitial pneumonia (NSIP) was found in 56 patients.

#### 3.1.4. Histopathological Patterns

A lung tissue biopsy was performed on 159 patients, mainly by video-assisted thoracoscopy (VATS) (n = 123); a transbronchial lung biopsy (TBLB) was performed on 36 patients (non-IPF diseases (HP, sarcoidosis, etc.) were usually suspected, and then the results of the TBBL were not taken into account). A histopathological UIP pattern was found in 15 patients, a probable UIP pattern was found in 5, and alternative diagnoses were made in 68 patients. Among the cases with an alternative disease, 16 patients (24.2%) had a histopathological pattern of HP, and 6 patients (8.8%) had morphological features of CTDs.

### 3.2. Diagnosis of IPF

A central MDD review was used in 48 patients. Of these, IPF was diagnosed in 16 patients (34.8%), probable IPF was diagnosed in 5 patients (10.9%). Diagnostic criteria of an alternative disease were found in 25 patients (54.3%). Unclassifiable IIP was diagnosed in one patient based on a low clinical probability of IPF, undetermined for UIP patterns in lung HRCT and lung biopsies, while not having any signs of an alternative disease.

### 3.3. Treatment

Data on their pharmacological treatments were available for 948 patients. Systemic steroids were administered most often (in 406 of 948 patients; 42.8%), followed by proton pump inhibitors (120 of 948 patients; 12.7%), and N-acetylcysteine (92 of 948 patients; 9.7%). During the study period, the rate of administration of these agents reduced by 15.7%, 6.2%, and 1.5%, respectively. Antifibrotic drugs were administered to 90 of 948 patients (9.5%) at the time of entering the registry; 61 patients were treated with nintedanib and 29 patients were treated with pirfenidone. During the study period, the number of patients treated with these antifibrotics increased by 5.4% and 6.2%, respectively (Figure 6). The type and frequency of adverse effects (AEs) were consistent with the known safety profiles of pirfenidone and nintedanib, including gastrointestinal events, which were the most frequent treatment-emergent AEs.

Data on non-pharmacological treatment were available for 893 patients. Non-pharmacological treatment was administered to 198 of 948 patients (20.9%), including long-term oxygen therapy in 151 of 948 patients (15.9%), pulmonary rehabilitation in 16 of 948 patients (1.7%), and non-invasive ventilation in 11 of 948 patients (1.2%).

### 3.4. Mortality

Data on their vital status were available for 1329 patients. Of them, 132 patients (9.9%) died during the study. The most frequent causes of death were worsening or exacerbation of IPF (40.9%) and worsening of chronic respiratory failure (18.2%). Other causes of death included pulmonary embolism (n = 5), pneumonia (n = 1), lung carcinoma (n = 1), spontaneous pneumothorax (n = 1), myocardial infarction (n = 1), and acute heart failure (n = 1).

### 3.5. Subgroup Analysis

The length of the disease (LOD), from the appearance of the first symptom to entering the registry, was known for 985 patients; the LOD was > 6 months in 904 (91.8%) and < 6 months in 81 patients (8.2%).

Patients with an LOD ≤ 6 months and > 6 months did not differ in mean age (64.6 ± 10.7 and 65.8 ± 10.6 years, respectively; *p* = 0.39), smoking history (28.3 ± 16.0 and 30.5 ± 19.0 pack-years, respectively; *p* = 0.59) and comorbidities. The patients with an LOD ≤ 6 months were predominantly males (67.9% vs. 56.1%; *p* = 0.04), were older at the time of the disease onset (65.5 ± 10.6 vs. 60.8 ± 11.4 years; *p* = 0.0006), and had a significantly lower LOD at the time of entering the registry (0.3 ± 0.1 vs. 3.8 ± 3.2 years; *p* < 0.0001) compared to patients with an LOD > 6 months. Also, patients with an LOD ≤ 6 months had lower prevalences of dyspnea (86.4% vs. 96.6%; *p* = 0.00001), cough (65.4% vs. 80.9%; *p* = 0.001), fatigue (48.1% vs. 61.1%; *p* = 0.02), and finger clubbing (21.0% vs. 1.6%; *p* = 0.002) compared to patients with an LOD > 6 months. The prevalence of other signs and symptoms did not differ significantly between the groups.

At the time of inclusion in the registry, the mean MRC dyspnea rating was lower (1.5 (1.0–3.0) vs. 2.0 (2.0–3.0); *p* = 0.0001), and oxygen saturation SpO2 at rest was higher (96% (93–97%) vs. 94% (91–96%); *p* = 0.005), in the patients with an LOD ≤ 6 months compared to those with an LOD > 6 months. The lung function parameters (FVC and FEV1) were also higher in the patients with an LOD ≤ 6 months, though lung volumes (VC and TLC) and lung diffusion capacities did not differ significantly between the groups (Table 1).

Exercise tolerance was significantly better in patients with an LOD ≤ 6 months compared to those with an LOD >6 months (six-minute walk test (6-MWT) distance walked, 377.1 ± 150.5 m vs. 304.5 ± 142.0 m; *p* = 0.019, respectively). The SpO2 at baseline and at the end of the 6-MWT was also higher in the patients with the LOD ≤ 6 months (96% (95–97%) vs. 95% (93–97%); *p* = 0.018, and 94% (88–95%) vs. 89% (84–92%); *p* = 0.007, respectively). Desaturation during exertion was noted in 7 of 20 patients (35%) with an LOD ≤ 6 months and in 259 of 396 patients (65%) with the LOD > 6 months, for whom this information was available. Borg’s scale dyspnea did not differ significantly between the groups.

Patients with an LOD ≤ 6 months were treated with systemic steroids significantly less often compared to patients with an LOD > 6 months (8.6% vs. 43.4%; *p* = 0.000001).

## 4. Discussion

Our study is the first large epidemiological study on IPF in the Russian Federation. According to preliminary data, the prevalence and morbidity of IPF in the Russian Federation is 8 to 12 cases per 100,000 and 4 to 7 cases per 100,000, respectively [30]. Similar epidemiological data were published for USA and European countries [31]. There was an increase in the prevalence of diagnoses of IPF in Russia in 2016, when physicians were actively trained to participate in the registry, and in the subsequent two years of functioning of the registry. This demonstrates an improvement in the physicians’ knowledge of the diagnosis of IPF. The registry helps a practitioner to get opinions from an expert pathologist and an expert radiologist with experience in diagnosis of ILD in general and particularly of IPF. This is of particular importance for remote regions of the country not having a large tertiary care center.

Compared to most registries, working since 2011–2012, the Russian registry of IPF patients started in 2016. The Russian IPF patients were younger (65 years of age compared to 68.7–73.0 years of age for patients in other countries), but the length of the disease was similar. There were more active smokers in the Russian registry compared to other registries (12.6% vs. 1.7–9%), but the portion of patients who had never smoked was similar compared to other registries (42.7% vs. 27–47%) [10,12,15].

An interesting feature of our registry, which should be mentioned, is the slight predominance of women among patients with IPF (57.5%). This result is in disagreement with other registries or cohort studies [32]. However, this gender difference in IPF patients in our study is not unique. For example, a slight female predominance, or near equal gender distribution, were demonstrated in cohort studies from India (54% and 50%) [33,34], Saudi Arabia (51%), [35] and Pakistan (46%) [21]. On the one hand, heterogeneity of the patients in different registries should be regarded as one of registries’ strengths [9]. The variability in the characteristics of IPF patients from different parts of the world may be associated with genetic differences [36], environmental factors [37], additionally, differences in geographic and ethnic populations may also play a role. On the other hand, it is necessary to understand the limitations of our registry, which, in addition to patients with IPF, presumably also included patients with other fibrotic ILDs. It should also be added that in a recent national survey from Russia, the majority of patients with IPF were men (66%) [30]. Thus, further studies are needed to obtain a more accurate understanding of the characteristics of IPF patients in our country.

Rates of different lung function abnormalities did not differ between Russian patients with IPF and patients from other IPF registries. Of note, pulmonary function tests were used in the Russian Federation less often than in other countries (68% vs. 92–100%) [10,12,20]. Moreover, the rate of administration of systemic steroids was higher, and the rate of administration of antifibrotic agents was lower in the Russian Federation compared to other countries (42.8% vs. 8–30% and 9.5% vs. 22–78%, respectively) [10,12,20].

A comparison between the Russian registry of patients with IPF and IPF registries of other countries are given in Table 2.

One of the goals for development of the Russian registry of patients with IPF was to evaluate the need for antifibrotic drugs in the Russian Federation to plan for the burden of PF for our public health service. A post-hoc analysis showed that a significant portion of patients included in the registry had non-IPF fibrosing ILDs. According to this goal, the Expert Board of Russian Respiratory Society decided not to withdraw the patients with other diseases from the registry. In 2018, several sub-registries were formed out of the IPF registry, such as sub-registries for hypersensitivity pneumonitis, unclassified IIPs, other fibrosing ILDs, and a sub-registry of alternative diseases. Such heterogeneity of patients could explain some results, such as high blood eosinophils in 11.8% of patients, positive CTD serological markers in 9% of patients; a relatively low rate of UIP (17.5%) and probable/possible UIP patterns (34.2%) on HRCT; alternative pathohistological patterns in 68 of 159 patients (42.8%), and the low rate of confirmed diagnoses of IPF (20.8%).

A multidisciplinary approach within the registry allowed IPF patients to be provided with expensive antifibrotic drugs. Registries of patients with IPF with extended follow-up are necessary to evaluate the effectiveness and safety of these IPF treatments in the real-world setting, and could assist in the design of future studies with new experimental drugs for IPF (such as pamrevlumab, pentraxin-2, autotaxin inhibitors, galectin-3 inhibitors, phosphodiesterase 4 inhibitors, HSP90 inhibitors, NLRP3 inhibitors, etc.) [38,39,40]. 

The Russian registry of patients with IPF demonstrated a difference between patients with various LOD. Patients with an LOD ≤ 6 months had less dyspnea on daily activity and higher blood oxygen saturation both at rest and on exertion compared to the patients with more prolonged course of the disease. Surprisingly, pulmonary function did not differ significantly between these two groups: TLC and DLCO were similar in patients with different LODs, though FVC and FEV1 were significantly higher in patients with an LOD ≤ 6 months, both as absolute values and as %predictive. This discrepancy could be due to the different size of these groups. Nevertheless, the rate of IPF progression is defined independent of the annual rate of FVC decline [41]. Therefore, further analysis of subgroups with different LODs could provide important information on the IPF course in Russian patients and be used to evaluate the efficacy of antifibrotic therapy.

Behr et al. also analyzed, in the INSIGHT-IPF registry, the course of IPF in relation to the LOD [10]. Of 502 patients included in the registry, 26.7% of patients had an LOD < 6 months, and 73.3% had an LOD ≥ 6 months. In the German registry, patients with an LOD < 6 months were older and had more preserved DLCO compared to patients with a longer disease history (71.0 vs. 67.5 years and 37.7% pred vs. 17.6% pred, respectively). We did not find a significant difference between similar subgroups for these parameters, probably due to a high prevalence of patients with LODs > 6 months in our registry (904 vs. 81 patients). Notwithstanding, Russian patients with different LODs differed significantly in the severity of symptoms, physical tolerance in 6-MWT, and oxygen saturation. This difference was not found in the INSIGHT-IPF registry. This inconsistency could be due to the fact that the Russian registry improved early diagnosis of IPF in Russia, while German investigators explain their results by different periods of data collection, different survival, and other biases related to the inclusion of patients in the registry [10].

The experience of our registry confirmed once more that the differentiation between IPF and other fibrosing ILDs is extremely complex. In our registry, 43.3% of patients were previously exposed to various environmental hazards such as antigens of birds or livestock, and molds; 11.8% of patients had a high blood eosinophil count. These features require differentiation between IPF and chronic fibrosing hypersensitivity pneumonitis. Moreover, an HP histopathological pattern was diagnosed in 24.2% of patients who underwent lung tissue biopsy. Finally, 9.2% of patients had at least one increased serological marker of CTD, and 8.8% of patients had histopathological patterns consistent with CTDs. Similar findings were reported in other IPF registries. CTDs were diagnosed in 10% of patients in an Australian registry [13]; and 5.6% of patients from a Spanish registry had positive serological markers of CTD [17]. Morell et al. reported that 43% of patients with IPF, according to international guidelines, had a subsequent diagnosis of chronic hypersensitivity pneumonitis [42].

We should note that the number of patients treated with systemic steroids did reduce during the period of our study. We consider this to be caused by an improvement in physicians’ awareness of the negative effects of systemic steroids on IPF morbidity and mortality [30].

Non-pharmacological treatment of IPF, particularly pulmonary rehabilitation, is also of great importance. In a recently published systematic review and meta-analysis, Yu et al. have shown that pulmonary rehabilitation, primarily physical training, could improve daily physical activity and quality of life of patients with IPF [43]. Florian et al. demonstrated that pre-transplantation physical training could improve post-transplantation survival of patients with IPF [44]. Pulmonary rehabilitation involves 10% [17] to 23% of patients with IPF [Vasakova, 2018, unpublished] in Europe compared to 1.6% of patients in Russia.

In general, real-life trials are greatly beneficial. Particularly, a registry of IPF patients generally includes all patients with the disease being seen by the physician. This allows evaluation of epidemiology, morbidity and prevalence of IPF, diagnostic approaches, and treatment in various regions of the country.

Our study has several limitations. First, the results of our study are limited by factors that are inherent to any real practice registry. The information available for patients was based on individual clinical practices and not on the study protocol. Although we expected the questionnaire to be fully completed for each visit and every patient, some important data were missing. The findings submitted by physicians could contain misprints or technical errors. It was also not possible to control the quality of pulmonary function tests, exercise tests, etc. As the experience of maintaining a registry increased, we increased the requirements of the quality of data submitted to the registry, but a significant body of data accumulated during the first years had been excluded from the analysis for technical reasons and misprints. Second, another limitation of our study is the fact that the diagnosis of IPF was not always the result of MDD. The recent guidelines established the crucial role of MDD in the diagnosis of IPF [1,2,3]. MDD might provide a specific diagnosis discordant with pre-MDD diagnosis, and be particularly valuable in the diagnosis of non-IPF ILD [45]. Although MDD was standard procedure in many expert centers participating in the registry when discussing the diagnosis and management of IPF, in a number of respiratory centers the access to an MDD was not possible (most often in facilities with an insufficient number of ILDs patients), and the diagnosis of IPF was based on the opinion of a treating pulmonologist. Also, in some controversial cases, central MDD reviews were conducted. Third, the information on the presented causes of mortality also needs to be clarified. For example, among the causes of death in patients with IPF in our registry, lung cancer was practically absent (the most common causes were “worsening” or “exacerbation” of IPF and “worsening of chronic respiratory failure”). However, according to epidemiological studies, the percentage of deaths due to lung cancer in IPF patients is about 2.5% [46]. A possible explanation for this inconsistency with the known data is the inherent limitations of presenting information on the causes of death in the registry. It is highly likely that deaths due to lung cancer in IPF patients were labeled as “worsening”. And fourth, since this was not a clinical trial, we cannot make accurate comparisons between different treatment groups, as in randomized trials. Therefore, more accurate and long-term data will be revealed in the future.

## 5. Conclusions

The Russian registry of patients with IPF is the first epidemiological study of IPF in the Russian Federation. This work resulted in an improvement in the diagnosis of IPF, including earlier diagnosis, and implementation of antifibrotic drugs in the clinical practice. We demonstrated the advantages of early diagnosis of IPF and highlighted the insufficient involvement of patients with IPF on pulmonary rehabilitation programs. Further studied are needed to follow-up on the clinical course of IPF, to find prognostic markers of IPF, and to analyze the long-term efficacy of antifibrotic therapy.

## Figures and Tables

**Figure 1 life-13-00435-f001:**
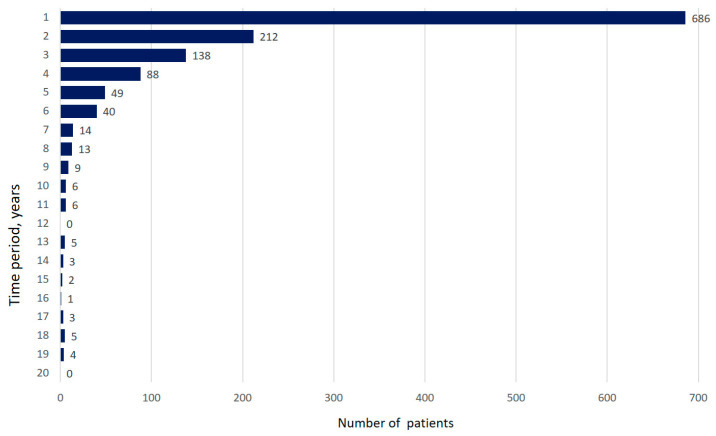
Time between onset of first symptoms and diagnosis of IPF (years). (Data are presented as absolute number of patients).

**Figure 2 life-13-00435-f002:**
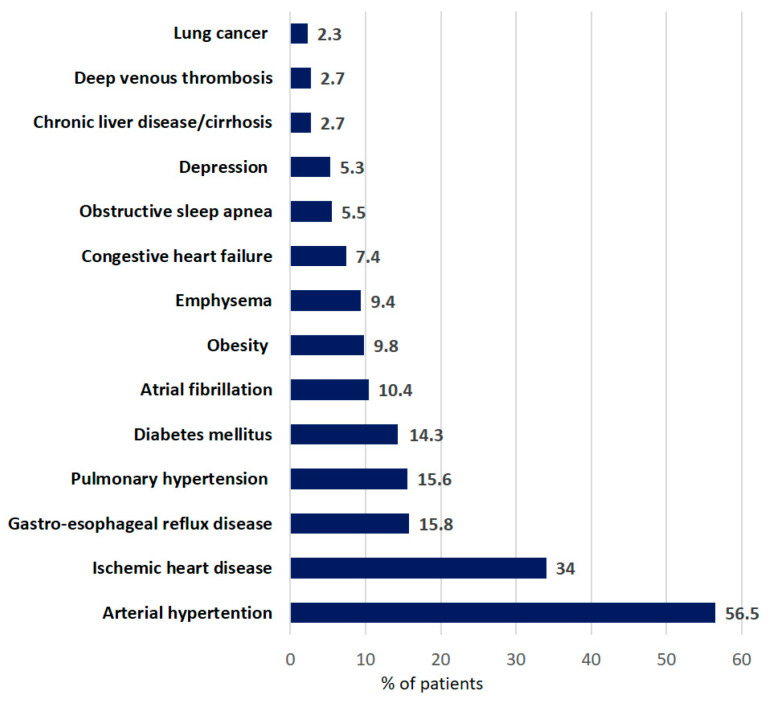
Distribution of comorbidities in patients with IPF at baseline. (Data are presented as percentage of all patients).

**Figure 3 life-13-00435-f003:**
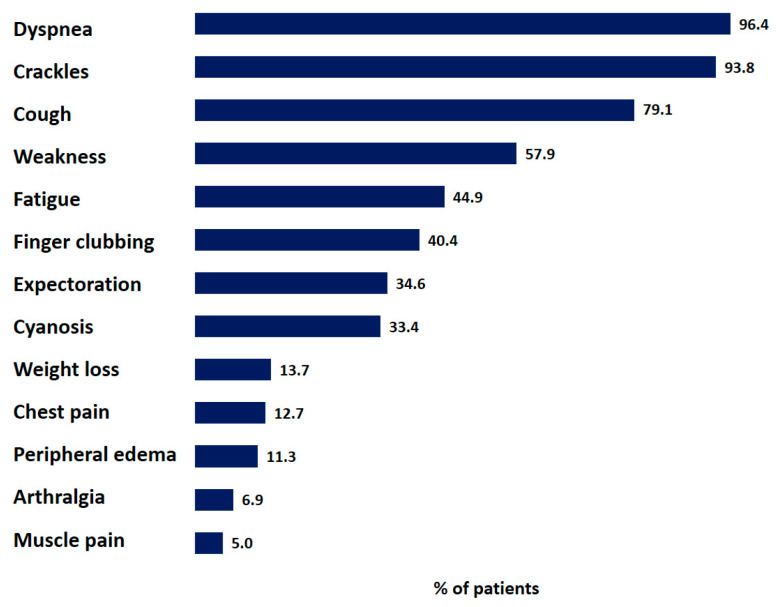
Distribution of symptoms and signs of IPF patients. Data are presented as percentage of all patients with reported symptoms or signs.

**Figure 4 life-13-00435-f004:**
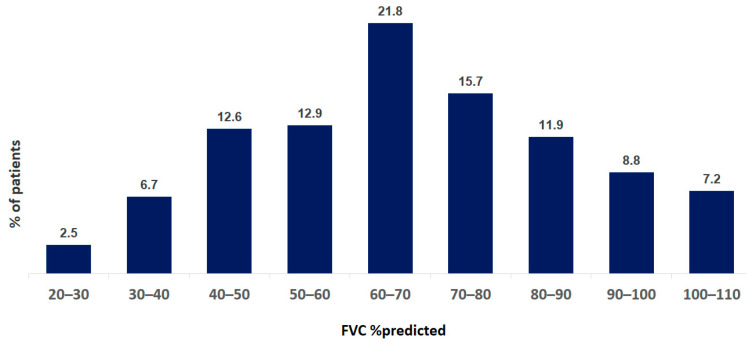
Distribution of FVC data at baseline of IPF patients. Data are presented as percentage of all patients.

**Figure 5 life-13-00435-f005:**
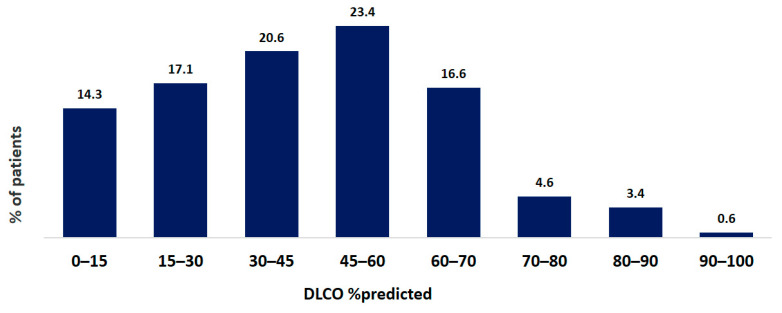
Distribution of DLCO data at baseline of IPF patients. Data are presented as percentage of all patients.

**Figure 6 life-13-00435-f006:**
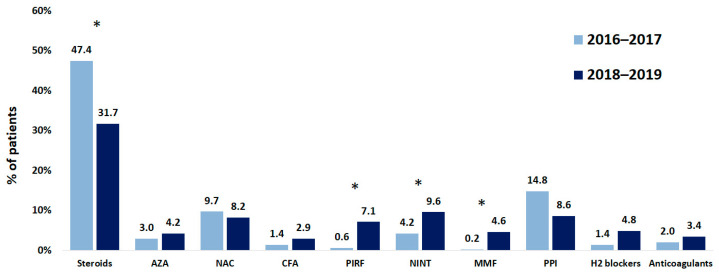
Changes in therapies of patients with IPF while registry was running. NAC, N-acetylcysteine; CFA, cyclophosphamide; AZA, azathioprine; PIRF, pirfenidone; NINT, nintedanib; MMF, mycophenolate mofetil; PPI, proton pump inhibitors. * *p* < 0.01.

**Table 1 life-13-00435-t001:** Lung function in patients with the length of the disease ≤ 6 months and > 6 months at inclusion in the registry.

Parameters	Patients with the LOD ≤ 6 Months	Patients with the LOD > 6 Months	*p*
Number of patients, n	81	904	
FVC, % pred.	78.9 ± 23.7	69.3 ± 22.6	0.003
FVC, L	2.79 ± 1.08	2.36 ± 0.97	0.008
FEV_1_, %pred.	81.8 ± 21.9	72.8 ± 22.9	0.003
FEV_1_, L	2.31 ± 0.80	1.96 ± 0.73	0.003
FEV_1_/FVC, %	85.1 ± 8.7	84.7 ± 11.4	0.54
TLC, %pred.	74.6 ± 19.6	71.3 ± 18.6	0.32
TLC, L	4.79 ± 1.77	4.34 ± 1.40	0.24
VC, %pred.	84.3 ± 23.8	75.0 ± 21.0	0.06
VC, L	2.99 ± 1.13	2.62 ± 0.96	0.15
DLCO, %pred.	50.9 ± 16.5	46.9 ± 17.5	0.33
DLCO/V_A_, %pred.	75.8 ± 14.9	67.8 ± 18.4	0.13

Abbreviations: FVC, forced vital capacity; FEV_1_, forced expiratory volume for 1 s; TLC, total lung capacity; VC, vital capacity; DLCO, diffusion lung capacity for carbon monoxide; VA, alveolar volume.

**Table 2 life-13-00435-t002:** Comparison between Russian IPF registry and IPF registries of other countries.

	Russian Registry	INSIGHT-IPF [10]	SEPAR [17]	Australian IPF Registry [13]	PROOF [14]	IPF-PRO [18]	Finland IPF Registry [19]	EMPIRE (Czech Part) [11,16]	Latin American Registry of IPF [22]	Korea IPF Cohort Registry [23]
A year of starting the registry	2016	2012	2012	2012	2013	2014	2011	2012	2017	2008
Patients’ age, years	64.4 ± 10.7	68.7 ± 9.4	70.2 ± 9.2	70.9 ± 8.5	69.6 ± 8.6	70 (65–75) *	73.0 ± 9.0	67 (50–82) #	71.9 ± 8.3	67.4 ± 9.3
Males/females, %	42.5/57.5	77.9/22.1	80.8/19.2	67.7/32.3	76.9/23.1	74.9/25.1	65.1/34.9	70/30	74.7/25.3	76.1/23.9
Active smokers, %	12.6	1.7	9.0	71.7	6.5	68.4	7	53	2.4	12.9
Ex-smokers, %	44.7	60.4	63.7		66.8		48		52.3	58.1
Never smoking, %	42.7	38.0	27.3	28.3	26.7	31.6	45	47	45.3	29.0
The length of disease before making the diagnosis	19.8 ± 28.3 months	46.9 ± 52.5 months	20.4 ± 21.4 months	NA	281 days	14 (7–19) * months	575 ± 799 days	12 (2–48) # months	12 (6–24) months	NA
FVC, % pred.	69.7 ± 22.3	72.2± 20.6	77.6± 19.4	81.0± 21.7	80.6± 19.9	69.6 (60.1–9.95) *	80.2± 18.2	80,0 (48.7–16.3) #	70.9 ± 19.8	74.6 ± 17.5
TLC, % pred.	71.5 ± 18.8	70.2 ± 21.1	72.5 ± 16.5	NA	NA	NA	NA	NA	NA	NA
DLco, % pred.	46.2 ± 17.1	35.5 ± 15.5	48.5± 17.7	48.4± 16.7	46.9± 13.8	41.7 (32.2–50.1) *	55.6 ± 16.5	45.6 (21.3–72.3) #	53.7 ± 45.4	63.6 ± 22.0
6-MWD, m	305 ± 141	268 ± 200	423.5 ± 110.4	420 ± 129	426.2 ± 130.8	NA	NA	NA	380.0 ± 135.8	399 ± 182
HRCT, %	97.8	90.2	99.2	93	97.5	NA	NA	100	NA	~100
PFTs, %	68.1	98.8	98.3	92	NA	NA	NA	100		~100
Lung biopsy, %	16.2	34.1	NA	13	30.3	30	NA	NA	16.3	31.9
Patients treated with SCS before entering the registry, %	42.8	26.1	17.8	46	8.3	NA	NA	29.7 ^&^	4.2	13.7
Patients treated with antifibrotics before entering the registry, %	9.5	44.2	69.4	23	69.3	54	26	78 ^&^	72	34.1

Data are presented as mean ± SD; *, median (25th–75th percentile); #, median (interquartile range); NA, data are not available. TLC, total lung capacity; DLCO, diffusion lung capacity for carbon monoxide; 6-MWD, six-minute walk distance; HRCT, high resolution computed tomography; CSC, corticosteroids, PFTs, pulmonary function test. ^&^ Vasakova, unpublished, 2018.

## Data Availability

Not applicable.

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
