# Peer review of "Russian Registry of Idiopathic Pulmonary Fibrosis: Clinical Features, Treatment Management, and Outcomes"

_life, 2023, doi:10.3390/life13020435_

Round 1

Reviewer 1 Report

Chikina et al in the present study made an effort to understand Idiopathic Pulmonary Fibrosis in a clinical setting.

Here are comments 

1. I would like the authors to explain in detail the immunological aspect and progression of the disease in the introduction.

2. If you could provide examples of the HCRT scans for control and infected patients?

3. Just curious to know if there are any side effects of these antifibrotic drugs? 

Overall, it is a preliminary study to understand the disease in a clinical setting which further can open up different avenues to explore.

Author Response

We thank the reviewer for their careful review of our manuscript and for their useful and interesting comments and suggestions. We offer our sincere gratitude to the reviewer for taking the time to extensively review the manuscript. 

Please see attached a point-by-point response to your concerns and our revised manuscript. 

All edits and new information included in the revised version of the manuscript are highlighted in yellow.

All edits and new information included in the revised version of the manuscript are highlighted in yellow.

Reviewer 2 Report

The manuscript of Dr. Avdeev and co-authors is well-written and presents for the first time the official clinical data and the epidemiology of idiopathic pulmonary fibrosis in Russia. The comparison with other countries looks very interesting. There are some comments that authors should consider before the manuscript is accepted.

1. In Table 2, the authors provide data on gender differences among patients. The presented data are very different from the world average. It is well known that IPF is a sex-dependent disease, and men are much more likely to develop pulmonary fibrosis than women. The authors should suggest in the Discussion part, why in Russia the registration of women with IPF is higher than that of men.

2. The amount of quantitative information is quite wide and complete, but the manuscript has only three figures. I would strongly suggest adding some additional figures, to make the manuscript easier to read.

3. The discussion should be expanded a little bit to include data on experimental drugs for IPF, such as ATX inhibitors, HSP90 inhibitors, NLRP3 inhibitors, and so on, and prospects for their use in the Russian Federation.

Minor:

1. The description of figures 1 and 2 should be more detailed. 

2. I would recommend that you do not specify the full number of federation regions in the methods, since readers from all countries will not agree with the specified number. Just leave the number of regions included in the statistics (73).

Author Response

We thank the reviewer for their careful review of our manuscript and for their useful and interesting comments and suggestions. We offer our sincere gratitude to the reviewer for taking the time to extensively review the manuscript. 

All edits and new information included in the revised version of the manuscript are highlighted in yellow.

Reviewer 3 Report

The introduction could be improved and described better the importance of early diagnosis and the importance of the National Registry.

In the Method maybe could be useful to describe how you trained

the doctors, and how they are stimulated to introduce the data. In the discussion chapter, the newest article should be cited, not until 2019.

I

Author Response

We thank the reviewer for their careful review of our manuscript and for their useful and interesting comments and suggestions. We offer our sincere gratitude to the reviewer for taking the time to extensively review the manuscript. 

Please see attached a point-by-point response to your concerns and our revised manuscript. All
edits and new information included in the revised version of the manuscript are highlighted in yellow.

Reviewer 4 Report

The authors reported the registry study of IPF in Russia. I believe that this registry study will play a significant role in the future development of IPF practice in their country by identifying the characteristics and problems of IPF clinical practice in Russia. I found this study to be very beneficial to the Russian public. However, I have some comments that I think need to be reconsidered.

Major point

1.

3.1.4. histopathological patterns. This paper described 36 cases of TBLB for which a pathological evaluation was performed. However, the current international guidelines (REF #1) state that TBLB is not useful for histopathological evaluation of IPF except for genetic testing. It is necessary to inform the current situation in which TBLB was performed for IPF, and it should be mentioned that TBLB was performed in 36 cases. However, I believe these 36 cases should be excluded from consideration for subsequent description (evaluation of tissue pattern and pathology diagnosis). Alternatively, the evaluation at TBLB should be listed separately.

2.

3.2. Diagnosis of IPF. There is one patient with MDD diagnosed as Unclassifiable IIP in this registry. The reason why this patient was classified as Unclassifiable IIP should be stated. If possible, please describe which description in the international guidelines (REF #1) this patient met to be classified as Unclassifiable IIP.

3.

3.4 Mortality. I think it is a major feature of this study that there are fewer deaths due to lung cancer than in other country registries. The causes of this should be discussed. Please consider adding a statement about this.

Minor point

4.

Introduction references a treatment for IPF, but I think it is inadequate; major studies on pirfenidone and nintedanib are needed. Please consider adding the following references.

l  King TE Jr, et al. N Engl J Med. 2014; 370: 2083-92.

l  Richeldi L, et al. N Engl J Med. 2014; 370: 2071-82.

5.

In this study, MDD was not performed in all cases for the diagnosis of IPF. The current international guidelines state that MDD is required in all cases for the diagnosis of IPF. I think this should be mentioned as a limitation.

6.

HRCT and histopathological pattern in the guideline is "Indeterminate for UIP", not undetermined UIP. Please correct.

Author Response

We thank the reviewer for their careful review of our manuscript and for their useful and interesting comments and suggestions. We offer our sincere gratitude to the reviewer for taking the time to extensively review the manuscript. 

Round 2

Reviewer 2 Report

The manuscript was significantly improved and could be accepted in its present form